# Controllable field-free switching of perpendicular magnetization through bulk spin-orbit torque in symmetry-broken ferromagnetic films

Xuejie Xie[1,3], Xiaonan Zhao[1,3], Yanan Dong[1], Xianlin Qu[2], Kun Zheng [2], Xiaodong Han [2], Xiang Han[1], Yibo Fan[1], Lihui Bai[1], Yanxue Chen[1], Youyong Dai[1], Yufeng Tian [1✉] & Shishen Yan [1✉]

Programmable magnetic field-free manipulation of perpendicular magnetization switching is essential for the development of ultralow-power spintronic devices. However, the magnetization in a centrosymmetric single-layer ferromagnetic film cannot be switched directly by passing an electrical current in itself. Here, we demonstrate a repeatable bulk spin-orbit torque (SOT) switching of the perpendicularly magnetized CoPt alloy single-layer films by introducing a composition gradient in the thickness direction to break the inversion symmetry. Experimental results reveal that the bulk SOT-induced effective field on the domain walls leads to the domain walls motion and magnetization switching. Moreover, magnetic field-free perpendicular magnetization switching caused by SOT and its switching polarity (clockwise or counterclockwise) can be reversibly controlled in the IrMn/Co/Ru/CoPt heterojunctions based on the exchange bias and interlayer exchange coupling. This unique composition gradient approach accompanied with electrically controllable SOT magnetization switching provides a promising strategy to access energy-efficient control of memory and logic devices.

[1] School of Physics, State Key Laboratory of Crystal Materials, Shandong University, Jinan, China. [2] Beijing Key Lab of Microstructure and Property of Solids, Institute of Microstructure and Properties of Advanced Material, Beijing University of Technology, Beijing, China. [3]These authors contributed equally: Xuejie Xie, Xiaonan Zhao. ✉email: yftian@sdu.edu.cn; shishenyan@sdu.edu.cn

Breaking the structural symmetries of nanomagnetic systems allows for physics that is forbidden in symmetric systems, promoting unique fundamental interactions between electrical current and magnetization. Especially, spin–orbit coupling (SOC) combined with inversion symmetry breaking is a core ingredient to achieve electrical manipulation of perpendicular magnetization using spin–orbit torque (SOT) towards ultralow-power, nonvolatile memory, and logic devices[1–19]. Such SOT-induced magnetization switching mostly exists either in heterostructures consisting of a ferromagnet and a heavy metal with strong SOC (see refs. [3–7,20,21]), or in bulk non-centrosymmetric single-layer films such as CuMnAs, Mn₂Au, and (Ga,Mn)As with globally or locally broken inversion symmetry[8–10]. Up to now, remarkable progresses have been made, such as low-power magnetization switching[4,5,11], fast domain-wall motion in synthetic antiferromagnetic racetracks[14,15], giant spin transfer torque generated by a topological insulator[16–18], and so on. However, with the symmetry consideration of many common ferromagnetic films with a centrosymmetric space group, such as Fe, Co, Ni, FeCo, FePt, and CoPt, they themselves cannot have the bulk spin–orbit coupling induced non-equilibrium spin polarization to switch the magnetization[22,23], which greatly limit their practical applications in the SOT-based functional devices. Therefore, how to introduce the SOT-switching capability into a common single-layer ferromagnetic film itself is a great challenge.

In order to realize the SOT-switching in the common single-layer ferromagnetic film itself, here, we propose a CoPt composition gradient approach to break the inversion symmetry in otherwise bulk centrosymmetric ferromagnetic alloys. The SOT-switching of the single-layer ferromagnetic film has one obvious advantage that without the need of introducing any extra heavy metal layer to produce SOT effect, the SOT-switching of the single-layer magnetic film can be directly added into the existing devices based on magnetic tunneling junctions, magnetic spin valves, and exchange biased systems to realize a new class of memory and logic functions by SOT effect.

On the other hand, in order to operate nonvolatile memory and logic devices by electrical writing and electrical reading, magnetic field-free SOT induced magnetization switching has been realized by the employment of exchange bias[24,25], interlayer exchange coupling[26], lateral wedge structures[27], ferroelectric control[28], geometrical engineering[29], and spin current manipulations[30–32]. However, the switching polarity (clockwise or counterclockwise) of the SOT induced magnetization switching is usually difficult to be reversibly controlled by external manners. Here in order to further realize a controllable field-free SOT-switching, we introduce the CoPt composition gradient film into IrMn/Co/Ru/CoPt heterojunctions with the synthetic antiferromagnetic coupling and exchange bias, where the switching polarity can be reversibly controlled by electrical current.

In this work, we not only propose a composition gradient approach to break the inversion symmetry in otherwise bulk centrosymmetric ferromagnetic alloys, which make magnetization switching through bulk spin–orbit coupling possible, but also demonstrate a controllable field-free perpendicular magnetization switching by utilizing the exchange bias and interlayer exchange coupling. As a proof-of-principle experimental example, we demonstrate room temperature SOT switching of the perpendicularly magnetized CoPt alloy single-layer films by introducing an artificial composition gradient in the thickness direction to break the inversion symmetry. It is revealed that the SOT-induced effective field on the domain walls (DWs) leads to the DWs motion and magnetization switching. In addition, field-free perpendicular magnetization switching is obtained in the IrMn/Co/Ru/CoPt heterojunctions, and its switching polarity can be reversibly controlled. This composition gradient method, which

enables the selective and independent control of the magnetization of a single-layer ferromagnetic film, together with controllable field-free SOT switching holds promising applications for electrically controllable magnetic memory and logic devices.

## Results

**Characterization of the composition gradient CoPt alloy.** The CoPt ferromagnetic alloy single-layer films with an artificial composition gradient in the thickness direction, which are designed as the nominal multilayered structure of Pt(0.7)/Co(0.3)/Pt(0.5)/Co(0.5)/Pt(0.3)/Co(1) (thickness in nanometer), are deposited on Si–SiO₂ substrate/Ru buffer layer by using magnetron sputtering at room temperature. Then the deposited films are processed into Hall-bar structures with a 70-μm-long channel along the x-axis and 5-μm in width, as shown in Fig. 1a. The high-resolution transmission electron microscopy (HRTEM) image in Fig. 1b indicates that a high-quality smooth and continuous CoPt alloy film of 3.3 nm in thickness is obtained. The high-angle annular dark-field (HAADF) image in Fig. 1c and the HRTEM image in Supplementary Fig. 1 also indicate that CoPt layer in the IrMn/Co/Ru/CoPt exchange biased sample is a nanocrystal single-layer alloy film, rather than the nominal multilayered structure of Pt(0.7)/Co(0.3)/Pt(0.5)/Co(0.5)/Pt(0.3)/Co(1). Moreover, the lattice constant of CoPt(111) is slightly smaller than that of the pure Pt(111) as shown in Supplementary Fig. 2, which is consistent with previous reports[33].

In order to further confirm the formation of CoPt alloy in our studied systems, we have prepared a control sample of Ru(2)/[Pt(8)/Co(6)]₃/Ru(2) with relatively thick Pt and Co layers, and a clear multilayered structure can be observed in the elemental mapping and the HAADF images as shown in Supplementary Fig. 3. However, it is observed that a very thin region about 1.2 nm appears at the Pt/Co interfaces, which has formed the CoPt alloy due to atomic diffusion during the sputtering deposition as confirmed by the line scanning of the energy-dispersive X-ray spectroscopy (EDS) (Supplementary Fig. 3f). Hence, in our studied systems when very thin Pt and Co layers were alternatively sputtered for several times, CoPt alloy single-layer instead of multilayer is synthesized. By contrast, thick Pt and Co layers form Pt/Co multilayers with interfacial alloying.

On the other hand, the elemental mapping of EDS in Fig. 1d and the HAADF image in Fig. 1e reveal that the Ru/IrMn/Co/Ru/CoPt/MgO heterojunction still keeps a good multilayered structure due to relatively weak interdiffusion at the interfaces. Despite the interfacial diffusion, Fig. 1 and Supplementary Fig. 3c indicate that the thin Ru layer of 0.8 nm is still a separate and continuous sublayer but without sharp interface, which is consistent with the strong antiferromagnetic interlayer exchange coupling in the Ru/IrMn/Co/Ru/CoPt/MgO heterojunction.

Now, we focus on the composition gradient of the CoPt alloy layer. The line scanning of EDS results of Ru/IrMn/Co/Ru/CoPt/MgO heterojunction is shown in the inset of Fig. 1e. We can see that the Co atomic fraction monotonically increases along the thickness direction from the bottom to the top of the CoPt alloy layer, while the Pt atomic fraction monotonically decreases. It is clear that the composition gradient is along the thickness direction of CoPt alloy layer. However, due to the spread of EDS data, the variation of the composition in our designed samples may be not as continuous as the EDS data, and a more continuous composition gradient can be prepared by continuous deposition.

In the CoPt composition gradient single-layer alloy films, the magnetization versus magnetic field curves (shown in Supplementary Fig. 4a) indicate that the easy magnetization axis of the CoPt films is perpendicular to the film plane. Here, the average saturation

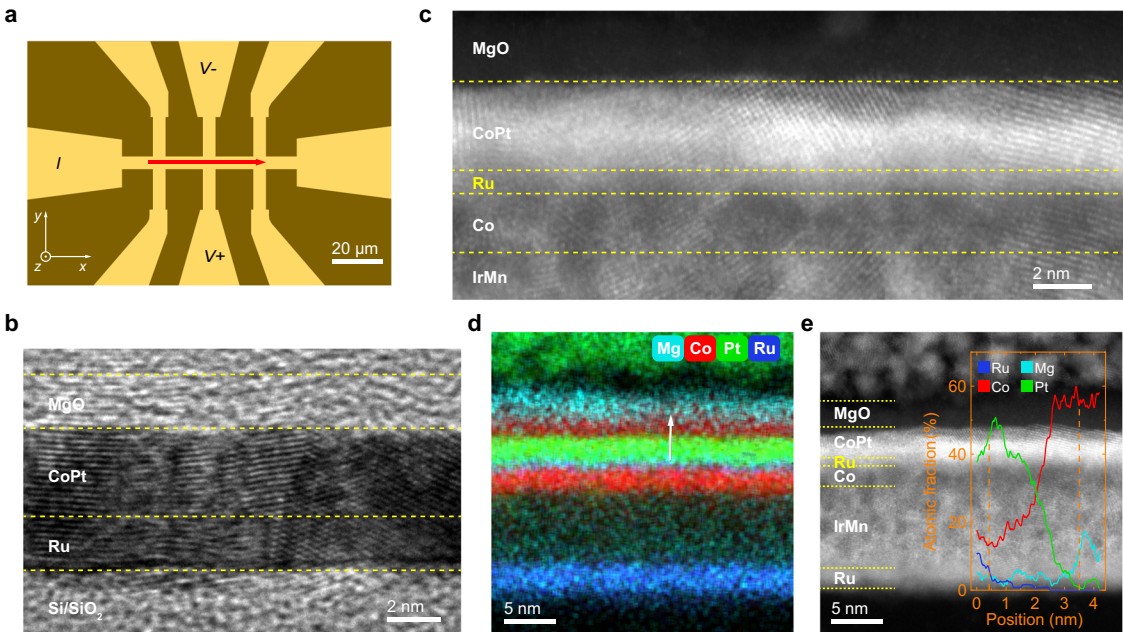

**Fig. 1 Microstructure and composition characterization. a** An optical micrograph of the fabricated 70 μm × 5 μm Hall bar. **b** The high resolution cross-section transmission electron microscopy image of the CoPt alloy single-layer film with the nominal multilayered structure of Pt(0.7)/Co(0.3)/Pt(0.5)/Co(0.5)/Pt(0.3)/Co(1), which is grownn on Ru/Si buffer/substrates and capped by MgO protective layer. **c** the enlarged high-angle annular dark-field (HAADF) images, **d** the elemental mapping, **e** the HAADF images of the Ru(2)/IrMn(8)/Co(2)/Ru(0.8)/CoPt(3.3)/MgO(2) heterojunction. Inset of **e** shows the line scanning of the EDS results marked by a white arrow in **d**. It is clear that the designed nominal multilayered structure of Pt(0.7)/Co(0.3)/Pt(0.5)/Co(0.5)/Pt(0.3)/Co(1) has formed the single-layer CoPt alloy with an artificial composition gradient in the thickness direction. In contrast, the Ru(2)/IrMn(8)/Co(2)/Ru(0.8)/CoPt(3.3)/MgO(2) heterojunction still keeps a good multilayered structure.

magnetization of the 3.3 nm CoPt single-layer films is about 250 emu cm$^{-3}$ at room temperature. The measured anomalous Hall resistance $R_H$ in Supplementary Fig. 4b, which is proportional to the average vertical component of the CoPt magnetization $M_z$ (see ref. [34]), also shows the establishment of perpendicular magnetic anisotropy.

**Current-induced SOT switching and chiral domain wall movement**. Now we report the reversible electrical current induced magnetization switching for the perpendicularly magnetized CoPt single-layer films. In Fig. 2a, with a small magnetic field along x-axis ($H_x = \pm 100$ Oe), a hysteric magnetization switching between $M_z > 0$ ($R_H > 0$) and $M_z < 0$ ($R_H < 0$) is observed by sweeping pulsed current along the x-axis. The switching polarity (clockwise or counterclockwise) of magnetization switching reverses when the magnetic field reverses[3,35], which is the typical feature of SOT-induced magnetization switching. Under a positive field $H_x = +100$ Oe, the positive current favors $M_z < 0$, which is independent of the initial magnetization states (Supplementary Fig. 5). On the other hand, the significant current induced magnetization switching is only observed around $\theta = 90°$ ($\theta = 90°$ represents the magnetic field along the x-axis), and the switching polarity of magnetization switching is not affected by small geometric tilting ($\triangle \theta \approx \pm 4°$ around $\theta = 90°$ at $H_x = \pm 100$ Oe), as shown in Supplementary Fig. 6.

The current-induced switching loops measured under various $H_x$ indicate that the clockwise (counterclockwise) switching loops are obtained when positive (negative) $H_x$ is applied, as shown in Supplementary Fig. 7. Increase in the amplitude of $H_x$ gradually squeezes the SOT-switching loops and reduces the critical switching current density[3,36–38]. The observed SOT-switching has good thermal stability (up to 300 °C) as shown in the Supplementary Fig. 8. In addition, we have grown two series of samples with different composition gradients, i.e., one series of

samples have positive Co composition gradients (Co composition increases from the bottom to top of the CoPt layer), and the other series of samples have negative Co composition gradients. The SOT induced magnetization switching has been observed in all these samples as shown in Supplementary Figs. 9 and 10, indicating good repeatability and film-to-film reliability of the observed SOT-switching. More interestingly, not only the polarity of magnetization switching, but also the Dzyaloshinskii–Moriya interaction (DMI) effective magnetic field, and the chirality of the DWs reverse when the composition gradient becomes opposite (it will be discussed below), highlighting the critical role played by the composition gradient in the magnetization switching.

To further clarify the importance of the composition gradient to the SOT-switching, two kinds of control samples are prepared. In the Pt(2)/Co(0.5)/Pt(0.5)/Co(0.5)/Pt(0.5)/Co(0.5)/Pt(2) sample with symmetric structure and no commposition gradient, it is found that although the perpendicular magnetization can be easily switched by the external magnetic field, it cannot be switched by the applied in-plane electrical current due to the symmetric structure, as shown in Supplementary Fig. 11. On the other hand, in the Ru(2)/Pt(3.5)/[Pt(0.5)/Co(0.5)]$_8$/MgO sample with no composition gradient but still the deposition induced interfacial asymmetry, relatively thick [Pt(0.5)/Co(0.5)]$_8$ nominal structure is designed to reveal the possible bulk SOT effect. However, the magnetization switching can not be observed within the maximum applied electrical current density of $4.5 \times 10^7$ A cm$^{-2}$ as shown in Supplementary Fig. 12. Therefore, the [Co/Pt]$_8$ sample indicates that only interfacial asymmetry itself without the composition gradient can not lead to enough strong spin–orbit torque to switch the magnetization of a thick magnetic layer with a large coercivity. All the above experimental results together unambiguously indicate that the SOT induced deterministic switching of magnetization can be attributed to introducing the composition gradient in the thickness direction to break the inversion symmetry.

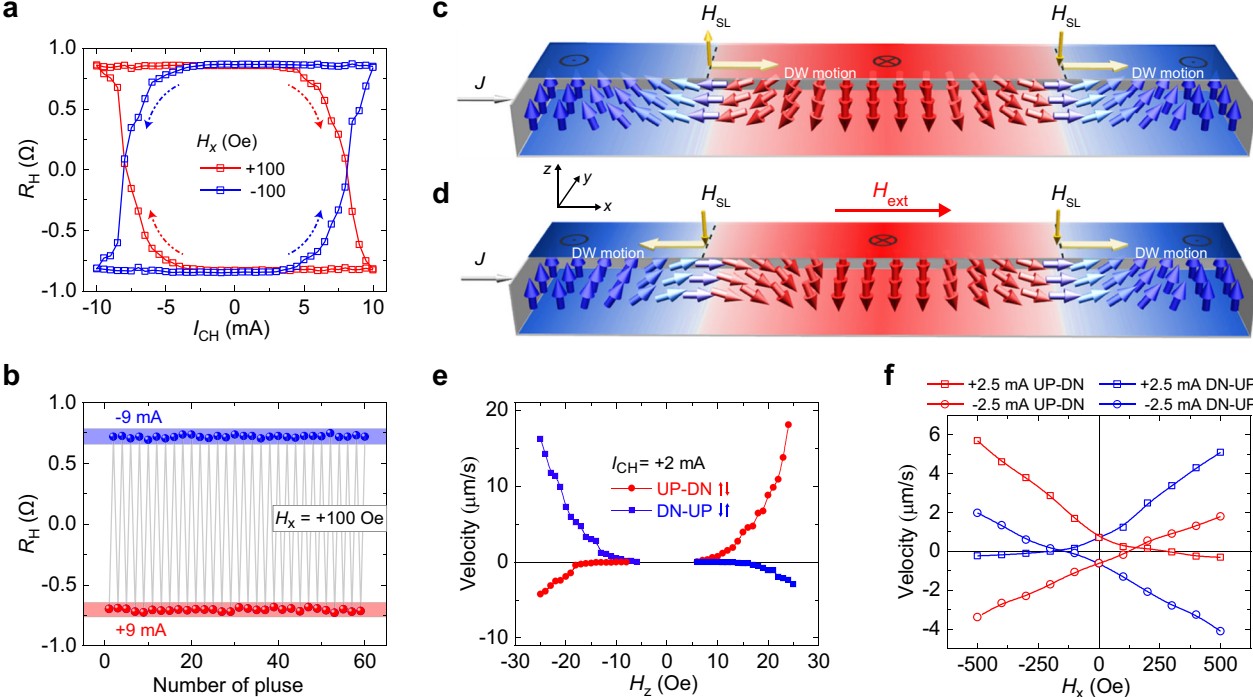

**Fig. 2 Current-induced magnetization switching and chiral domain wall displacement. a** Current-induced magnetization switching measured under $H_x = \pm 100$ Oe. The switching polarity is marked by the colored arrows. **b** Reversible magnetization switching induced by two opposite pulsed current of $\pm 9$ mA at $H_x = +100$ Oe. **c** The schematic illustration of the left-handed chiral Néel DWs without the application of in-plane magnetic field, where the effective field $H_{SL}$ associated with a Slonczewski-like torque drives adjacent up-to-down and down-to-up DWs moving towards the same direction with respect to the current. **d** The corresponding situation when a magnetic field is applied along $+\hat{x}$ direction, where the left-handed up-to-down DWs changes into the right-handed up-to-down DWs, resulting a reversion of both the $H_{SL}$ and the DWs motion direction. **e** Up-to-down (abbreviated as UP-DN, red circles) and down-to-up (abbreviated as DN-UP, blue squares) domain wall velocity as a function of out-of-plane magnetic field $H_z$ under $I_{CH} = +2$ mA. **f** Domain wall velocity versus in-plane magnetic field $H_x$. Red and blue symbols represent up-to-down and down-to-up DWs, respectively. Square and circular symbols correspond to positive and negative currents, respectively.

When the bulk spin–orbit coupling keeps constant and plays dominant role, in principle there is no limit to the thickness of switchable magnetic films. However, as the Co composition $\delta$ in $Co_\delta Pt_{1-\delta}$ ($\delta$ is at %) alloy film linearly increases from $\delta_1$ to $\delta_2$ with the magnetic layer thickness $t$, the Co composition gradient varies with the thickness as $(\delta_2 - \delta_1)/t$. Thus, due to small composition gradient in thick single magnetic layer, the SOT effective magnetic field may become too small to switch the magnetization. Further experiments indicate that a single CoPt composition gradient magnetic film thicker than 7 nm cannot be switched by the SOT effect. In fact, a thicker CoPt magnetic film can be obtained by periodically depositing the CoPt composition gradient layer. As a demonstration, we prepared a magnetic film of the total thickness up to 9.9 nm with the nominal structure of $[Pt0.7/Co0.3/Pt0.5/Co0.5/Pt0.3/Co0.7/MgO0.3]_3$, where a very thin MgO layer of 0.3 nm (should be not continuous) was inserted to enhance the perpendicular magnetic anisotropy, but not to decouple the ferromagnetic interlayer coupling. In this structure, the $[Pt0.7/Co0.3/Pt0.5/Co0.5/Pt0.3/Co0.7]$ forms CoPt alloy single-layer and adjacent CoPt alloy layers show strong ferromagnetic interlayer coupling, since the very thin MgO layer of 0.3 nm can not separate the adjacent CoPt alloy layers. Therefore, this sample can show magnetic behavior just like a single ferromagnetic film (Supplementary Fig. 13). The clear SOT induced magnetization switching indicates that the bulk spin–orbit torque can switch the magnetization of thick artificial magnetic structures. It is worthy to mention that the thickness of switchable CoPt alloy films in the range of 3.3–9.9 nm is comparable to those observed in other bulk spin-orbit torque

systems[12,20,21,39], but contrasts the very thin magnetic layer usually about 1 nm in the Pt/Co bilayers[3,5,6]. The switching of thick magnetic layer with perpendicular magnetic anisotropy is beneficial for practical spintronic applications[20,21,40].

In order to reveal the mechanism of SOT induced magnetization switching in the CoPt alloy composition gradient films, the magnetic domains were directly observed by magnetic optical Kerr microscope. As shown in Supplementary Figs. 14 and 15, we can clearly see that the DWs nucleation and propagation dominate the magnetization switching. It is noticed that the critical current of the electrical current line is smaller than that at the junction, which probably due to the current shunting through the arms of the Hall cross[24]. Therefore, the critical current is defined as the value of $I_{CH}$ at which $R_H$ changes sign in $R_H - I_{CH}$ loops (in Fig. 2a) measured by anomalous Hall effect in the Hall cross, which corresponds to the magnetization switching at the junction observed by MOKE images. Thus, a critical switching current of 7.5 mA in Fig. 2a, corresponding to a current density of $2.8 \times 10^7$ A cm$^{-2}$, can be regarded as the up-limit of the current density for switching in our sample.

Based on the DWs propagation observed in Supplementary Figs. 14 and 15, Fig. 2c, d show the working principle of the SOT switching of perpendicular magnetization in the composition gradient single-layer films. It is believed that the Dzyaloshinskii–Moriya interaction[35,41] stabilizes the homochiral Néel DWs (here left-handed Néel DWs with up-to-down and down-to-up configurations) in the CoPt films, as shown in Fig. 2c. For the composition gradient along the thickness direction ($z$-axis), we can assume that the bulk symmetry-broken is along the

$z$-axis direction, which show the same type of symmetry-broken as interface/surface Rashba effect. Here, the bulk Rashba spin–orbit coupling Hamiltonian of an conducting electron can be described by $\hat{H}_R = \alpha(\hat{z} \times \hat{k}) \cdot \hat{\sigma}$, where $\hat{\sigma}$ is the operator of Pauli-spin matrices and $\hat{k}$ is the wave vector of the conducting electron (along the opposite direction of electrical current $J$), $\hat{z}$ is the unit vector of the $z$-axis (the film growth direction), and $\alpha$ is the Rashba strength parameter. When an electrical current $J$ is injected along $+\hat{x}$ direction (electrons move along $-\hat{x}$ direction), the effective magnetic field felt by the conducting electron is $\alpha(\hat{z} \times \hat{k})$, along $-\hat{y}$ direction. Thus a pure spin current with spin polarization $\hat{\sigma}$ along $+\hat{y}$ direction is produced to transport along the $+\hat{z}$ direction due to the bulk Rashba spin–orbit coupling. The spin current with spin polarization $\hat{\sigma}$ along $+\hat{y}$ direction can produce a field-like effective magnetic field along $-\hat{y}$ direction on the local magnetization vector $M$ of magnetic domains or domain walls in the ferromagnetic layer, which do not depend on the magnetization $M$. On the other hand, the spin current with spin polarization $\hat{\sigma}$ along $+\hat{y}$ direction can produce a Slonczewski-like effective magnetic field $H_{SL}$ (proportional to $M \times (\hat{z} \times \hat{k})$) on the left-handed up-to-down Néel DWs, as shown in Fig. 2c, which manifests itself with an out-of-plane component ($z$-axis) since there exists a component of the magnetization along the $-\hat{x}$ direction within the left-handed up-to-down Néel DWs (see refs. [35,42]). The effective field $H_{SL}$ can drive both the up-to-down and down-to-up DWs to move along the same direction of $+\hat{x}$, but it can not lead to the deterministic switching of magnetization without a bias magnetic field. However, if a bias magnetic field is applied along the $+\hat{x}$ direction, as shown in Fig. 2d, the up-to-down and down-to-up DWs move along the opposite direction of $x$-axis, which can reverse the magnetization.

It is worthy to mention that the DWs motion depends on the chirality of the DWs, the electrical current, and the magnetic field. From the magnetic domain evolution with time as shown in Supplementary Figs. 14 and 15, we can obtain the domain wall velocity as a function of the applied magnetic field and/or the electrical current, as shown in Fig. 2e, f. Figure 2e shows the experimental results of up-to-down and down-to-up domain wall velocity as a function of out-of-plane magnetic field $H_z$ under a fixed electrical current $I_{CH} = +2\,\text{mA}$. For the given positive magnetic field $H_z$, the up-to-down DWs feel a total magnetic field of $H_z + H_{SL}$ and show the positive (in $+\hat{x}$ direction) and large velocity, but the down-to-up DWs feel a total magnetic field of $H_z - H_{SL}$ and show the negative and small velocity. On the other hand, for the given negative magnetic field, the up-to-down DWs show the negative and small velocity, but the down-to-up DWs show the positive and large velocity. This phenomenon is one of the key features of the movement of left-handed chiral Néel walls, which can be schematically explained by Fig. 2c.

Figure 2f shows up-to-down and down-to-up domain wall velocity as a function of in-plane magnetic field $H_x$ under $I_{CH} = \pm 2.5\,\text{mA}$. We can see that the up-to-down domain wall velocity becomes zero around $H_x = 155\,\text{Oe}$, which can be explained by the external magnetic field $H_x$ canceling the internal DMI effective magnetic field ($-155\,\text{Oe}$) in the left-handed chiral Néel walls with up-to-down domain wall configuration[42]. For the same reason, the internal DMI effective magnetic field becomes opposite ($155\,\text{Oe}$) in the down-to-up domain wall configuration of the left-handed chiral Néel walls. Therefore, Fig. 2e, f convincingly give the experimental evidences of the DMI effective magnetic field which cause the left-handed chiral Néel walls in our CoPt single-layer films. On the other hand, for the left-handed chiral Néel domain walls, under the same applied electrical current (such as $+2.5\,\text{mA}$), the velocity of the up-to-down domain walls at $H_x$ is equal to that of the down-to-up

domain walls at $-H_x$. This degeneracy (or symmetry) is expected for the left-handed chiral Néel domain walls with the Slonczewski-like torque and the Dzyaloshinskii–Moriya interactions. By contrast, we found that CoPt films with negative Co composition gradient has the right-handed chiral Néel DW and the DMI effective magnetic field in the right-handed down-to-up DW is $-70\,\text{Oe}$, as shown in Supplementary Fig. 10e.

**Quantitative evaluation of the SOT effective field.** In this section, we quantitatively investigate the SOT effective field which act on the domain walls during the magnetization switching. Figure 3a shows the current induced shift of the out-of-plane hysteresis loops at a fixed bias field $H_x = +600\,\text{Oe}$ and $I = \pm 4\,\text{mA}$. The opposite loop shift along the $H_z$ axis under opposite electrical current indicates the existence of current induced effective field $H_{eff}^z$ ($H_{SL}$ in the above) due to the Slonczewski-like torque[43]. By plotting the switching fields for both up-to-down ($H_{SW}^{UP-DN}$) and down-to-up ($H_{SW}^{DN-UP}$) transitions as a function of applied electrical current, a switching phase diagram is obtained as shown in Fig. 3c, d, where the linear contribution (marked by black triangles) originates from the current-induced effective field ($\propto I$) and the nonlinear contribution comes from the Joule heating ($\propto I^2$) induced coercivity reduction[43]. The estimated current-induced effective field $H_{eff}^z = -\frac{H_{SW}^{UP-DN} + H_{SW}^{DN-UP}}{2} = -H_{shift}^z$ is about ~9.5 Oe per $10^7\,\text{A cm}^{-2}$, and its dependence on the applied electrical current is summarized in Fig. 3b. It can be seen that the slope of $H_{eff}^z/I$ reverses when the polarity of $H_x$ reverses, which can be well explained by the DWs motion scenario. Upon application of an in-plane bias field $H_x$ along $+\hat{x}$ direction, the magnetic moments in the middle of Néel walls realign parallel to $H_x$ as shown in Fig. 2d. In this case, when the electrical current is injected along $+\hat{x}$ direction, the effective field $H_{eff}^z$ (or $H_{SL}$) for both up-to-down and down-to-up DWs points along $-\hat{z}$ direction and therefore facilitates the expansion of domains with $-M_z$ magnetization, and finally magnetization reverses to $-\hat{z}$ direction in the whole sample, which can well explain the magnetization switching in Figs. 2a and 3a.

We also quantitatively analyze the longitudinal and transverse components of the SOT effective field on single magnetic domains rather than the domain walls by first and second harmonic measurements[44,45]. By taking into account the contribution from the planar Hall effect as shown in Supplementary Fig. 16, the longitudinal ($\Delta H_L$, Slonczewski-like, along $\pm\hat{x}$) and the transverse ($\Delta H_T$, field-like, along $\pm\hat{y}$) components of the effective field on one single magnetic domain are $\Delta H_L \approx 5.0\,\text{Oe}$ per $10^7\,\text{A cm}^{-2}$, and $\Delta H_T \approx 3.0\,\text{Oe}$ per $10^7\,\text{A cm}^{-2}$, respectively. From the SOT-induced longitudinal effective field, we can extract the spin Hall angle $\theta_{SH} \approx +0.011$ in our CoPt alloy films, which is much smaller than $\theta_{SH} \approx +0.08$ of Pt layer[46]. However, without considering the planar Hall effect, $H_L \approx 4.5\,\text{Oe}$ per $10^7\,\text{A cm}^{-2}$, $H_T \approx 2.2\,\text{Oe}$ per $10^7\,\text{A cm}^{-2}$, and effective spin Hall angle $\theta_{SH} \approx 0.010$ are obtained, as shown in Supplementary Fig. 17. Since the planar Hall effect is very small as compared with anomalous Hall effect, the effective field-like or damping-like field measured by first and second harmonic measurements only show a slight change after considering the correction from the planar Hall effect. On the other hand, it is very interesting to notice that the Slonczewski-like spin–orbit effective field $H_{eff}^z$ ($H_{eff}^z \approx 9.5\,\text{Oe}$ per $10^7\,\text{A cm}^{-2}$), that acts on the magnetic Néel domain walls is obviously bigger than that acting on the magnetic domains ($\Delta H_L \approx 5.0\,\text{Oe}$ per $10^7\,\text{A cm}^{-2}$). This is because Slonczewski-like spin–orbit effective field is related to the local magnetization vector, which is quite different in the magnetic domains and domain walls.

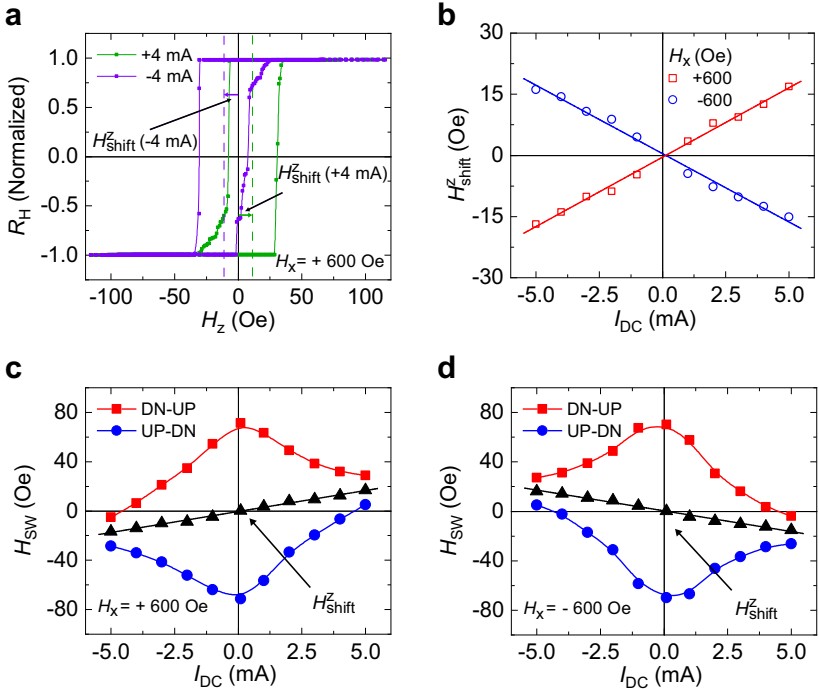

**Fig. 3 Evaluation of the z component of the spin–orbit effective field on the magnetic Néel DWs. a** The normalized out-of-plane Hall curves measured under a fixed in-plane magnetic field of $H_x = +600$ Oe and opposite d.c. currents of $\pm 4$ mA. **b** The $H_{shift}^z$ as a function of $I_{DC}$ under different bias fields. The solid lines are linear fittings to the data. **c**, **d** The switching fields $H_{SW}$ for down-to-up (DN-UP, red squares) and up-to-down (UP-DN, blue circles) magnetization reversals as a function of $I_{DC}$ with $H_x = +600$ Oe and $H_x = -600$ Oe, respectively.

With the SOT effective field in mind, we further unveil the mechanism of the electrical current induced magnetization switching. First, it is easy to exclude the Oersted field produced by the electrical current, as it can not explain the observed domain wall motion[42]. Second, the electrical current induced adiabatic and non-adiabatic spin transfer torques on the domain walls are expected to push the domains to move along the electron drift direction, regardless of the polarity of the domains[47]. So we can exclude the adiabatic and non-adiabatic spin transfer torques. Third, the spin Hall effect from the nonmagnetic Pt layer can be excluded in our CoPt alloy film, because the spin Hall effect is significant only when the thickness of Pt layer is larger than its spin diffusion length 1.4 nm[48], usually in the range of 2–6 nm. In fact, there exists no separate continuous Pt layer in the CoPt alloy film although it is prepared by alternatively sputtering very thin Co and Pt layers. Fourth, the interfacial Rashba effect of the bottom and top interfaces of CoPt alloy film (it indeed exists) is also unlikely to dominate the observed SOT switching, because its interfacial nature usually makes it switch the magnetization of an ultrathin ferromagnetic layer with a thickness around 1.0 nm (see refs. [6,7,49]) rather than 3.3 nm in our case. Finally, the composition gradient scenario of the magnetization switching in our CoPt alloy film is further proved by the fact that the polarity of magnetization switching reverses when the composition gradient become opposite, as shown in Supplementary Figs. 9 and 10. Combining all the experimental results in Figs. 1–3, we come to the conclusion that CoPt alloy composition gradient produces the bulk symmetry-broken along the z-axis direction, and leads to the bulk Rashba spin–orbit coupling of an conducting electron, which further results in a net current-induced spin–orbit torque to switch the magnetization of the magnetic alloy film[50,51]. In fact, the composition gradient scenario of the magnetization switching in our CoPt alloy film is well supported by the experimental

observation that the Slonczewski-like effective magnetic field $H_{eff}^z$ on the DWs leads to the DWs motion and magnetization switching as shown in Figs. 2 and 3.

**Controllable field-free SOT switching**. From the application point of view, a controllable field-free SOT switching of perpendicular magnetization is highly desired for a flexible design of functional memory and logic devices. Here, we show that in the IrMn/Co/Ru/CoPt heterojunctions with the composition gradient CoPt layer, not only the field-free SOT-induced perpendicular magnetization switching, but also its switching polarity of the CoPt alloy layer can be reversibly manipulated by controlling the in-plane exchange bias field between IrMn and Co. In the IrMn/Co/Ru/CoPt heterojunctions, the top CoPt alloy layer has perpendicular anisotropy, but the bottom Co layer has in-plane anisotropy, as shown in Supplementary Fig. 18. The bottom Co layer and top CoPt alloy layer are antiferromagnetically exchange coupled via a thin Ru spacer of 0.8 nm. In addition, the direct exchange coupling between the bottom Co layer and antiferromagnetic IrMn layer leads to a significant exchange bias field, which is consistent with the shift of in-plane magnetization hysteresis loops (Supplementary Fig. 18). Figure 4 shows the current-induced magnetization switching curves of the IrMn/Co/Ru/CoPt heterojunctions measured at various $H_x$. The most significant feature here is the remarkable field-free SOT-induced magnetization switching of the top CoPt alloy layer. Another character is that the SOT switching polarity reverses when $H_x$ is around $+550$ Oe, $-60$ Oe, and $-550$ Oe. The switching polarity is determined by the direction of effective bias field $H_{eff} = H_x + H_{IEC}$ acting on the top CoPt alloy, where $H_x$ is the external applied magnetic field and $H_{IEC}$ is the antiferromagnetic interlayer exchange coupling field. Here, as discussed above, the SOT switching of the perpendicularly magnetized CoPt alloy is

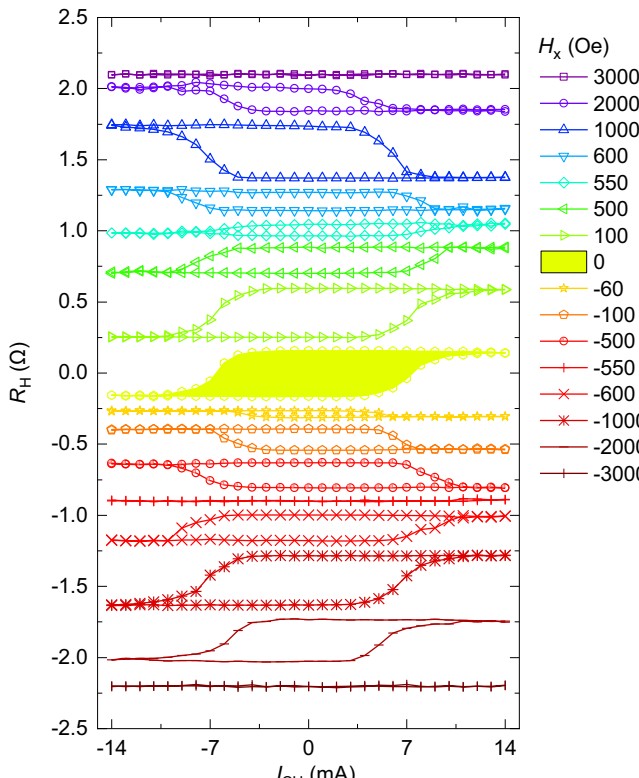

**Fig. 4 Field-free SOT switching based on exchange bias and interlayer exchange coupling.** The $R_H$–$I_{CH}$ loops measured under various in-plane magnetic fields $H_x$ ranging from −3000 Oe to 3000 Oe for the IrMn/Co/Ru/CoPt heterojunctions.

which is sensitive to the magnetization component along z-axis (mainly CoPt layer), as shown in Fig. 5b. It is clear that the IrMn/Co/Ru/CoPt heterojunctions show the field-free clockwise switching in the initial state. Then the initial state is reset by applying a single current pulse with various amplitude and direction under $H_x = +400$ Oe, and later the reset electrical current and magnetic field were turned off. Figure 5a indicates that the magnetic hysteresis loops shift to the negative magnetic field after reset with increasing the amplitude regardless of the direction of the current pulse. However, the exchange bias in Fig. 5a and the field-free magnetization switching in Fig. 5b remain unchanged (stable), even if the heterojunctions are reset under a magnetic field up to 5 Tesla without the simultaneous application of current pulse (not shown). This strongly suggests that the Joule heating plays a significant role in the reversal of the exchange bias field by heating the antiferromagnetic IrMn layer above its Néel temperature with the assistance of an external magnetic field[54,55]. It is clearly shown in Fig. 5c that the IrMn/Co/Ru/CoPt heterojunctions show field-free counterclockwise switching after reset at $I_{CH} = +14$ mA and $H_x = 400$ Oe, opposite to clockwise switching in the initial state. Therefore, by fine tuning of the in-plane exchange bias field through the application of a magnetic field and a large enough current pulse, the SOT switching polarity at zero magnetic field can be reversibly controlled as summarized in Supplementary Fig. 19. It is very important for practical applications that the field-free switching and controllable switching polarity provide more degrees of freedom for a flexible design of the functional memory and logic, since the IrMn/Co/Ru/CoPt heterostructure can serve as one block to be directly used in the conventional magnetic tunneling junctions, and magnetic spin valves to enable a new class of memory and logic devices by spin–orbit torque without external magnetic field.

## Discussion

Let us compare the main merits of some existing bulk SOT material systems[12,19–21,39,56,57], which may be promising for the practical applications in memory and logic devices. First, field-free bulk SOT switching of CoPt composition gradient film is demonstrated in IrMn/Co/Ru/CoPt heterojunctions, relying on synthetic antiferromagnetic interlayer coupling through a Ru spacer to an auxiliary Co layer exchange-biased by IrMn. Considering the fact that a remarkable SOT switching in our CoPt films can be observed at the external magnetic field of 100 Oe, the minimum interlayer exchange coupling field should be bigger than 100 Oe to achieve the field-free switching of the maximum magnetic layer thickness. In fact, if we choose a proper Ru layer thickness, we can obtain much big interlayer exchange coupling constant to realize the field-free switching of a relative thick magnetic layer. So this method may be extended to other bulk SOT material systems with perpendicular magnetic anisotropy such as L1$_0$ FePt single film[12], Co/Pd multilayers[20], Co/Pt multilayers[19,21,56,57], and CoTb amorphous single layer[39] to achieve field-free SOT switching, where previously an external magnetic field is needed to obtain the deterministic magnetization switching. Second, for our 3.3 nm CoPt alloy composition gradient films, the switching critical current density is about $2.8 \times 10^7$ A cm$^{-2}$, and the Slonczewski-like effective field is about 9.5 Oe per $10^7$ A cm$^{-2}$, which are comparable to those observed in heavy metal/ferromagnet bilayers[3], Co/Pt multilayers[19,21,56], 6 nm L1$_0$ FePt single film[12], and 10 nm CoTb amorphous single layer[39]. Third, the effective magnetic anisotropy energy density is, respectively, $K_{eff} = 4.82 \times 10^5$ erg cm$^{-3}$ and $K_{eff} = 3.97 \times 10^5$ erg cm$^{-3}$ for the 3.3 and 9.9 nm CoPt alloy. And the room temperature ($T = 293$ K) thermal stability factor $\Delta = K_{eff}V/k_BT$ increases from 77 for the 3.3 nm CoPt to 191 for the 9.9 nm CoPt, assuming that the magnetic volume $V$ of a bit of

dominated by the bulk SOT effect due to the broken centrosymmetry. However, spin–orbit torques from antiferromagnetic IrMn (see refs. [52,53]) could not play a dominant role on the top CoPt layer since the spin current decays quickly over 2 nm Co layer. For $H_x > 550$ Oe (or $H_x < -550$ Oe), the effective bias field $H_{eff} = H_x + H_{IEC}$ is dominated by $H_x$ and the IrMn/Co/Ru/CoPt heterojunctions show clockwise (or counterclockwise) switching in Fig. 4, which is the same as that of CoPt single-layer films in Fig. 2a. When $550$ Oe $> H_x > -60$ Oe, $H_{eff} = H_x + H_{IEC}$ is dominated by $H_{IEC}$ along $-\hat{x}$ direction, and the IrMn/Co/Ru/CoPt heterostructures show counterclockwise switching in Fig. 4. This implies that the antiferromagnetic interlayer exchange coupling field $H_{IEC}$ is about 550 Oe. However, the switching polarity becomes clockwise again when $-60$ Oe $> H_x > -550$ Oe, suggesting $H_{eff} = H_x + H_{IEC}$ is dominated by $H_{IEC}$ along $+\hat{x}$ direction. The magnetic measurements indicate that the IrMn/Co/Ru/CoPt heterojunctions switch as a whole from the (Co→CoPt↑) state at $H_x > -60$ Oe to the (Co←CoPt↓) state at $H_x < -60$ Oe. This implies that the coercivity of the IrMn/Co/Ru/CoPt heterojunctions is about 60 Oe.

Finally, we show how to control the switching polarity of field-free SOT-induced magnetization switching. The initial state is set by applying a positive current pulse of $I_{CH} = +14$ mA under $H_x = -400$ Oe, and then both the electrical current and magnetic field are removed. After that the magnetic hysteresis loops were measured by longitudinal magnetic optical Kerr effect, which is sensitive to the magnetization component along x-axis here (mainly the Co layer), as shown in Fig. 5a. It is clear that the magnetic hysteresis loops shift to the positive magnetic field in the initial state. Correspondingly, the field-free SOT-induced magnetization switching was measured by anomalous Hall effect,

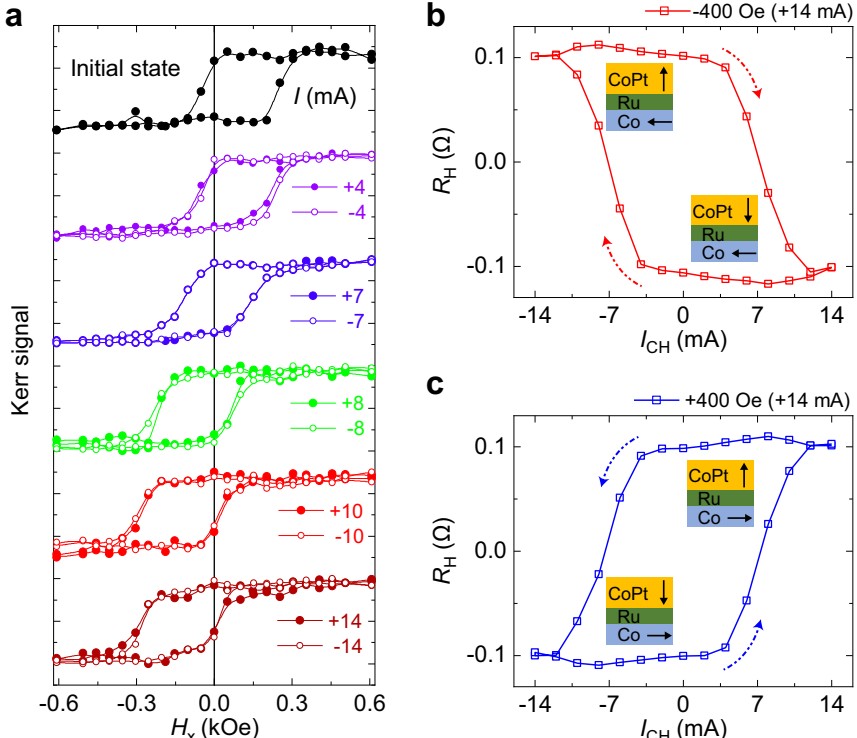

**Fig. 5 Controllable switching polarity of the field-free SOT switching. a** In-plane hysteresis loops of the IrMn/Co/Ru/CoPt heterojunctions measured by magnetic optical Kerr effect. The initial state is set by applying a positive current pulse of +14 mA under $H_x = -400$ Oe. Then the initial state is reset by a single current pulse with various amplitude and direction under $H_x = +400$ Oe. **b** The field-free clockwise switching in the initial state. **c** The field-free counterclockwise switching after reset by applying a positive current pulse of +14 mA under $H_x = 400$ Oe. The insets show the schematic magnetic configuration of Co layer and CoPt layer.

information is scaled down to 50 nm in diameter. The SOT efficiency, given by the effective areal anisotropy energy density divided by the switching current density $(K_{eff}t/J_{SW})$[21], also increases from $5.69 \times 10^{-16}$ J A$^{-1}$ for the 3.3 nm CoPt to $1.59 \times 10^{-15}$ J A$^{-1}$ for the 9.9 nm CoPt. The enhanced thermal stability and SOT efficiency are beneficial for the practical application. Still, as compared with the Co/Pt multilayers with high thermal stability and nanoscale scalability[21,56,57], the effective magnetic anisotropy energy density of our CoPt composition gradient film is relatively small, which will be further enhanced by optimal crystallization. However, it is clear that all the present study on the bulk SOT switching is mainly fundamental research. Therefore, at the present stage it is hard to say which bulk SOT material system is the most promising candidate for the practical applications in memory and logic devices.

Now let us discuss the relation between the composition gradient and the bulk SOT effect. The composition gradient itself can be greatly modulated by varying the layer thickness and layer number of Co and Pt layers, and correspondingly the SOT effective magnetic field can be effectively tuned as summarized in Supplementary Table 1. Qualitatively speaking, the spin Hall angle, the DMI effective magnetic field, the chirality of the DWs, the damping-like effective magnetic field, and the polarity of magnetization switching all reverses when the composition gradient becomes opposite. But it is hard to quantitatively compare these physical parameters between samples with different composition gradient. For example, the inset of 0.3 nm MgO in the 9.9 nm CoPt film (A4 sample in Supplementary Table 1) simultaneously enhance the coercivity and spin Hall angle as compared with the 3.3 nm CoPt film (A0 sample in Supplementary Table 1), and as a result they have similar critical switching current density. In fact, for most applications in

spintronic devices with the magnetic layer thickness usually less than 10 nm, it is easy to prepare the films with various composition gradient and showing efficient current-induced magnetization switching. Up to now, although the bulk spin–orbit torque has been found in a few magnetic single-layer films and multilayers[12,19–21,39,56,57], the mechanisms of structure/crystal inversion asymmetry are still controversial. However, rigorous theoretical calculations may be a good alternative approach to further verify the origin of bulk spin–orbit torque, and systematic experimental studies are required to obtain a quantitative relation between the SOT strength and the composition gradient, which is beyond the scope of this work.

In summary, within this study we have demonstrated the SOT switching of the perpendicularly magnetized CoPt single-layer ferromagnetic films by introducing the artificial composition gradient in the thickness direction to break the inversion symmetry. Moreover, a controllable field-free spin–orbit torque switching of the CoPt layer is obtained relying on the synthetic antiferromagnetic coupling and exchange bias in IrMn/Co/Ru/CoPt heterojunctions. Our findings may advance the practical applications of SOT systems in energy efficient memory and logic devices, since this simple composition gradient approach enables the selective electrical control of the magnetization switching of a single-layer ferromagnetic film.

## Methods

**Growth of composition gradient CoPt alloys**. The CoPt composition gradient alloys with the nominal multilayered structure of Pt(0.7)/Co(0.3)/Pt(0.5)/Co(0.5)/Pt(0.3)/Co(1) (thickness in nanometers) were deposited on a thermally oxidized Si (001) substrates coated with 300 nm thick SiO₂. A 2 nm Ru buffer layer with a weak spin–orbit coupling is used to avoid strong spin Hall effect of heavy metal while improving the crystal quality, and a 2 nm MgO capping layer was grown to protect the films from being oxidized by the atmosphere. On the other hand,

during the sputtering of MgO layer, partial oxygen atoms can diffuse into CoPt layer from the interface, which will further form the composition gradient of O atoms in the CoPt alloy and weaken the magnetization of CoPt alloy film. The Ru, Pt, and Co metal layers were deposited by d.c. magnetron sputtering at 3 mTorr Ar, and MgO was radiofrequency (RF) sputtered at 6 mTorr. The base pressure was better than $1 \times 10^{-5}$ mTorr. The deposition rates of Ru, Pt, Co and MgO respectively was 0.1136Å s$^{-1}$, 0.0835Å s$^{-1}$, 0.1377Å s$^{-1}$ and 0.0323Å s$^{-1}$, which had been calibrated by using X-ray reflectivity.

**Growth of IrMn/Co/Ru/CoPt heterojunctions**. The IrMn/Co/Ru/CoPt heterojunctions with the composition gradient CoPt alloy layer were prepared by sputtering, and have the thicknesses of Ru(2)/IrMn(8)/Co(2)/Ru(0.8)/CoPt(3.3), which were protected by 2 nm MgO as a cap layer. Then the heterojunctions were annealed at 300 °C under an in-plane magnetic field of 4000 Oe to promote the exchange bias between IrMn and Co.

**Microstructure and composition characterization**. The high resolution transmission electron microscopy (HRTEM) analysis, the high-angle annular dark-field (HAADF) imaging, and energy-dispersive X-ray spectroscopy (EDS) elemental analysis were performed with a FEI Titan Themis 60-300 microscope equipped with a probe spherical aberration (Cs) corrector operated at 300 kV. The EDS system is equipped with super-X detector and its energy resolution is 137 meV. The EDS line scanning and mapping results were performed under scanning transmission electron microscopy (STEM) mode.

**Magnetic optical Kerr effect (MOKE) measurements**. Magnetic domain images were observed by MOKE microscopy of an EVICO system. Both the time and the position of magnetic domain walls were extracted from the MOKE images. And the domain wall velocity was then calculated by using MOKE images over a certain time interval. The longitudinal Kerr signal of the IrMn/Co/Ru/CoPt heterojunctions shown in Fig. 5 was measured after the set/reset magnetic field and current pulses were removed.

**Magnetic and electrical properties measurements**. The magnetic hysteresis loops were measured using superconducting quantum interference device (SQUID, Quantum Design, MPMS-XL 7) and the typical sample size was 5 mm × 5 mm. For the electrical transport measurements, the films were patterned into conventional Hall bar structure with a channel width of 5 μm and length of 70 μm by using optical lithography and Ar-ion beam etching. Four-points Hall measurements were conducted using the Oxford physical property measurement system. The anomalous Hall resistance $R_H$ as a function of the external magnetic field was measured under a small continuous current of 0.1 mA. Current-induced SOT switching was measured by applying a 0.1 s current pulse with different amplitude (Keithley 6221) to the channel path of the Hall bar under a static in-plane magnetic field $H_x$. After each pulse (about 5 s), a small d.c. current $I_{dc}$ of 0.1 mA was applied and the Hall voltage $V_{dc}$ was recorded by Keithley 2182A. The Hall resistance $R_H = V_{dc}/I_{dc}$ was used to characterize the magnetization state. All measurements were conducted at room temperature.

**Perpendicular effective field acting on the DWs**. The out-of-plane anomalous Hall curves under different d.c. currents were measured by using Keithley 6221 and Keithley 2182A, while an in-plane bias magnetic field was applied along the current flow direction. Then we extracted the switching fields ($H_{SW}$) with magnetization from up-to-down (UP-DN) and down-to-up (DN-UP). The perpendicular effective field was defined as $H_{eff}^z = -\frac{H_{SW}^{UP-DN} + H_{SW}^{DN-UP}}{2} = -H_{shift}^z$.

**First and second harmonic measurements**. The current-induced effective fields on the magnetic domains were measured using a low-current excitation lock-in technique. A constant sinusoidal a.c. current $I_{ac}$ with a frequency of 547 Hz was provided by Keithley 6221. The in-phase first harmonic ($V_\omega$) and out-of-phase (90° off) second harmonic ($V_{2\omega}$) signals were recorded simultaneously by using two Stanford SR830 lock-in amplifiers, while sweeping the in-plane magnetic field directed transverse (along $\pm \hat{y}$) or parallel (along $\pm \hat{x}$) to the current flow. The remanent magnetization states $\pm M_z$ were obtained by applying a positive or negative out-of-plane magnetic field up to $\pm 4000$ Oe respectively, and then reducing to zero magnetic field. The effective transverse field $H_T$ and longitudinal field $H_L$, corresponding to field-like and damping-like (Slonczewski-like) torque, respectively, was obtained by fitting the experimental results according to the following equations:

$$H_{T(L)} = -2 \frac{\partial V_{2\omega}}{\partial H_{T(L)}} \bigg/ \frac{\partial^2 V_\omega}{\partial H^2_{T(L)}}.$$

The planar Hall effect (PHE) was measured by applying a 5 T in-plane magnetic field and rotating the sample around the vertical axis. Then the effective field-like

or damping-like field are corrected according to the following equation:

$$\Delta H_{L(T)} = \frac{H_{L(T)} \pm 2\xi H_{T(L)}}{1 - 4\xi^2}.$$

Here, the $\pm$ sign refers to the "up" and "down" magnetized configurations, while $\xi$ is the ratio between the planar ($R_{PHE}$) and anomalous ($R_{AHE}$) Hall resistances.

## Data availability

The authors declare that the main data supporting the findings of this study are available within the article and its Supplementary Information (Supplementary Figs. 1–19 and Supplementary Table 1). Extra data are available from the corresponding authors upon reasonable request.

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

## Acknowledgements

This work was supported by the National Natural Science Foundation of China (Grant Nos. 11774199, 11774016, and 51871112), the 111 Project B13029. The authors thank P. Shi and Q. K. Huang for their experimental support.

## Author contributions

Y.F.T. and S.S.Y. conceived and designed the experiments. Y.N.D., X.X.J., and X.H. fabricated the samples. X.L.Q., K.Z., and X.D.H. performed the TEM and EDS measurements. X.X.J., X.N.Z., and Y.B.F. performed the electrical and magnetic measurements. L.H.B., Y.X.C., and Y.Y.D. contribute to the data analysis. X.X.J., X.N.Z., Y.F.T., and S.S.Y. wrote the manuscript. All authors discussed the results, contributed to the data analysis and commented on the manuscript.

## Competing interests

The authors declare no competing interests.
