## [Peer Review File · Nature Communications]

Reviewers' Comments:

Reviewer #1:

Remarks to the Author:

The manuscript primarily reports on spin-orbit torque (SOT) electrical switching of the magnetization and its engineering in metallic multilayers prepared with a composition gradient, which constitutes an experimental realization of a new sub-category of SOT systems. The switching mechanisms are analyzed with a broad set of methods, providing consistent observations that demonstrate the mechanisms at play. Further, a current-reversible exchange-bias based on interlayer exchange coupling together with exchange-biasing by an antiferromagnet are demonstrated to work with the previously developed composition gradient magnetic layer. This notably allows for field-free switching of the magnetization by SOT with a switching polarity that can be independently set electrically with the reorientation of the antiferromagnet under field and Joule heating current.

The experimental results are very complete, soundly obtained, and provide strong support for the mechanisms of SOT switching and exchange biasing in the composition gradient Pt/Co multilayers under study. Key in the demonstration is that reversing the composition gradient reverses the torques. The manuscript is clear and well structured. In the rebuttal letter, the answers provided to previous criticism are satisfactory except several points (see below) that remain unclear. The manuscript will offer an even stronger demonstration if these are cured.

In terms of significance, going in the direction of bulk SOT switching has been demonstrated already, and the main figures of merit of the present CoPt composition gradient are equivalent to other bulk SOT systems, such as L10 FePt or CoTb, which likely limits the impact of the study. This is duly acknowledged in the manuscript. By contrast, the composition gradient here obtained from sputtered multilayers constitutes a new realization of bulk SOT switching which may prove more practical for combination with other functionalities. Maybe more important is indeed that this manuscript reports with solid details one of the very few demonstrations to date of a multilayer structure achieving an elaborated SOT engineering, here combining in an additive approach: a specific thickness-dependent modulation of magnetic properties to achieve SOT switching, interlayer exchange coupling through Ru, and exchange biasing with IrMn. This contributes to establish the maturity of the field of research, and in that sense this work constitutes a significant result that will probably find interest in the community.

Points to be clarified:

1. Control samples

a) The variations in the experimental conditions may weaken the quantitative comparison that is made between several switching loops. Why is H_x different between Fig. 2, sample A2, control samples Pt(2)/Co(0.5)/Pt(0.5)/Co(0.5)/Pt(0.5)/Co(0.5)/Pt(2) and [Pt(0.5)/Co(0.5)]₈ on the one side and samples A1, A3, [Pt0.7/Co0.3/Pt0.5/Co0.5/Pt0.3/Co0.7/Mg0.3]₃ on the other side? Does not this obscure the reasons for the absence of switching in some of them? I note that in Fig. 7a, all field values allow for switching at a similar current, but is that sufficient? Will this be true for all stacks, or could the minimum required H_x for switching be larger in some of them, e.g., Supp. Figs. 11 & 12? I suggest at least to add H_x in supplementary table I.

b) How is the magnetic state prepared before pulses in [Pt(0.5)/Co(0.5)]₈? The value of zero R_H does not fall on the H_z - R_H loop.

c) Significant fields are measured by second harmonic measurements in [Pt(0.5)/Co(0.5)]₈. As probably switching would occur at some point (beyond 4×10^7 A cm⁻²) for this control sample, all

I (mA) axes in the different no switching/switching loop figures could be converted in current density axes for improved clarity when comparing the different graphs. There is no reason to have either I or J instead of only J in the different graphs.

2. Gradient scaling with thickness

1379-382 "Bulk spin-orbit torque in magnetic films [...] is independent of the magnetic layer thickness, and therefore the critical current density and the SOT effective magnetic field can be similar for different magnetic layer thickness". While this can be true when the bulk SOT originates from structural asymmetry, I believe this statement to be misleading in the case of a gradient composition. The best achievable gradient reduces as $1/\text{thickness}$, and thus SOT effective magnetic field is limited by an amount scaling as $1/\text{thickness}$ as well (see, e.g., Liu et al., Phys. Rev. B 101, 220402(R) (2020)). This sentence needs to be corrected to clearly state this limit. Especially because, when stating "By contrast, the spin-orbit torque effective magnetic field based on spin Hall effect and/or interfacial Rashba effect strongly depends on the magnetic layer thickness", it actually also reduces exactly as $1/\text{thickness}$.

Additional comment:

The domain-wall (DW) motion depends indeed on the chirality of the DWs, the electrical current, and the magnetic field as reminded by the authors. They deduced left-handed chiral Néel walls in the multilayer Pt(0.7)/Co(0.3)/Pt(0.5)/Co(0.5)/Pt(0.3)/Co(1). Reversing the gradient of composition, this should reverse DMI and chirality of the DWs (at the same time that it reverses polarization sign of the spin currents). Out of curiosity, did the authors observe any clue of reversed DW chirality/DMI field sign in samples B1-B3 with opposite composition gradient? This would nicely correlate with the reversal of the SOT, on the phenomenological level.

Reviewer #2:

Remarks to the Author:

Review of manuscript resubmitted to nature communications

Review of corrections after previous response

This is a review of the modified manuscript, previously reviewed for a different Nature journal. Since previous reviews are on-record, I refrain from the standard full review of the paper's pros and cons.

In a nutshell, this paper is in the field of SOT switching of PMA thin films.

The 2 main claims are that 1) A magnetic "composition gradient" layer has broken inversion symmetry and thus enables SOT to arise directly in a PMA layer. 2) Demonstration of field free switching by coupling the PMA layer to a stable Co layer with in-plane magnetization.

Overall, the authors have done a good job incorporating previous comments from the reviewers, which has improved their paper.

To re-state my previous assessment. The ideas presented by the authors, of the SOT switching in a magnetic gradient layer are worthy of publication in Nature communications. The results support the claims in a satisfactory manner. This area of research is of interest to the scientific community and is also of relevance for applications. So, while the figure of merit is not (much) better than previously published results, bringing these concepts to the community is important, and maybe future work will also increase the FOM.

The second concept of field free switching is not related to the first. It is a proof-of-concept that it is possible to replace the external field with an antiferromagnetically coupled in-plane magnetic layer. It is demonstrated on the gradient layer, and the data and analysis are convincing.

Thus I support publication of this paper, after addressing a few final comments

First, the readability of the article has reduced. A good scientific editing is in place before publication.

An issue that I would like to have stated for the record – the EDS data of the TEM is always noisy, so there is always some spread of the results. I don't think that the gradient is a continuous gradient like the authors claim, but it is more of set-gradient that after EDS spread appears as a gradient. This does not change the results, but I would predict that a true gradient that is prepared by continuous deposition could have a different outcome.

Regarding the field free switching. This is an interface effect. So its usefulness will be reduced when a thicker FM layer is used, and thus it may be limited to thin layers. The authors should estimate what is the maximum thickness of the gradient layer (from their anisotropy energy analysis of the layer) that field free will work. Will it work for the 9nm film?

The results for the 9nm film are a bit strange, looking at the table in the supplementary. E.g., why is the theta so much larger for A4? Is it just because the thicker layer causes more of the current to run in the FM and not the Ru buffer layer? If so, then why is the critical switching current density not smaller? This actually is more reasonable, as the coercivity fields were larger, and the critical current was expected to also be larger in this case. Please comment.

Reviewer #3:

Remarks to the Author:

With the three rounds of the review process in Nature Electronics, the authors have made substantial revisions to this manuscript by conducting new experiments, improving the physics model, and implementing all the referees' comments. In addition, my comments in the last round of the review process have been carefully addressed and implemented in the manuscript accordingly. Although a quantitative model of bulk SOT remains to be established, its qualitative version interpreting the relation of bulk-SOT and composition gradient has been provided in accord with systematic experiments. I agree with the other referees that at this stage, nothing would affect the novelty of this work or prevent its publication.

As a fundamental study, this work will stimulate many further research proposals on this new bulk-SOT discovered in a magnetic system with a composition gradient and will help the researchers design more efficient magnetic storage. In my opinion, the current manuscript is already in a good shape in terms of convincing conclusions and clear statements of novelty and potential applications. Hence, I recommend this paper to be accepted in Nature Communications without major changes.

REVIEWER COMMENTS

Reviewer #1 (Remarks to the Author):

The manuscript primarily reports on spin-orbit torque (SOT) electrical switching of the magnetization and its engineering in metallic multilayers prepared with a composition gradient, which constitutes an experimental realization of a new sub-category of SOT systems. The switching mechanisms are analyzed with a broad set of methods, providing consistent observations that demonstrate the mechanisms at play. Further, a current-reversible exchange-bias based on interlayer exchange coupling together with exchange-biasing by an antiferromagnet are demonstrated to work with the previously developed composition gradient magnetic layer. This notably allows for field-free switching of the magnetization by SOT with a switching polarity that can be independently set electrically with the reorientation of the antiferromagnet under field and Joule heating current.

The experimental results are very complete, soundly obtained, and provide strong support for the mechanisms of SOT switching and exchange biasing in the composition gradient Pt/Co multilayers under study. Key in the demonstration is that reversing the composition gradient reverses the torques. The manuscript is clear and well structured. In the rebuttal letter, the answers provided to previous criticism are satisfactory except several points (see below) that remain unclear. The manuscript will offer an even stronger demonstration if these are cured.

In terms of significance, going in the direction of bulk SOT switching has been demonstrated already, and the main figures of merit of the present CoPt composition gradient are equivalent to other bulk SOT systems, such as L_{10} FePt or CoTb, which likely limits the impact of the study. This is duly acknowledged in the manuscript. By contrast, the composition gradient here obtained from sputtered multilayers constitutes a new realization of bulk SOT switching which may prove more practical for combination with other functionalities. Maybe more important is indeed that this manuscript reports with solid details one of the very few demonstrations to date of a multilayer structure achieving an elaborated SOT engineering, here combining in an additive approach: a specific thickness-dependent modulation of magnetic properties to achieve SOT switching, interlayer exchange coupling through Ru, and exchange biasing with IrMn. This contributes to establish the maturity of the field of research, and in that sense this work constitutes a significant result that will probably find interest in the community.

Reply: We would like to thank the reviewer for the careful reading of the manuscript and the constructive comments.

Points to be clarified:

1. Control samples

a) The variations in the experimental conditions may weaken the quantitative comparison that is made between several switching loops. Why is H_x different between

Fig. 2, sample A2, control samples Pt(2)/Co(0.5)/Pt(0.5)/Co(0.5)/Pt(0.5)/Co(0.5)/Pt(2) and [Pt(0.5)/Co(0.5)]₈ on the one side and samples A1, A3, [Pt0.7/Co0.3/Pt0.5/Co0.5/Pt0.3/Co0.7/MgO0.3]₃ on the other side? Does not this obscure the reasons for the absence of switching in some of them? I note that in Fig. 7a, all field values allow for switching at a similar current, but is that sufficient? Will this be true for all stacks, or could the minimum required H_x for switching be larger in some of them, e.g., Supp. Figs. 11 & 12? I suggest at least to add H_x in supplementary table I.

Reply: First of all, thanks for this suggestion, we have used the data with $H_x = \pm 100$ Oe for all the studied samples in the revised manuscript. And in the revised supplementary table 1, it has been pointed out that critical switching current density is obtained at $H_x = +100$ Oe.

Secondly, the SOT switching behavior is quite similar for all the studied samples measured under $H_x = 100$ Oe and $H_x = 200$ Oe. In fact, when the assistant magnetic field H_x is larger than 50 Oe, a remarkable SOT switching is observed for all the studied composition gradient single-layer films. On the contrary for the control samples without composition gradient (Supp. Figs. 11 & 12), no sign of remarkable SOT switching has been found even with $H_x = 1000$ Oe as shown in Reply-Figure 1.

Reply-Figure 1. **a.** The Hall resistance measured under the out-of-plane ($\theta = 0^\circ$) magnetic field (black curve) and the SOT-switching (red and blue curves) results of the C1 control sample [Pt(2)/Co(0.5)/Pt(0.5)/Co(0.5)/Pt(0.5)/Co(0.5)/Pt(2)]. **b.** The corresponding results for the C2 control sample [Ru(2)/Pt(3.5)/[Pt(0.5)/Co(0.5)]₈/MgO]. Both samples have been set to the demagnetization states before the SOT-switching measurements, and R_H keeps near zero if no magnetization switching occurs.

b) How is the magnetic state prepared before pulses in [Pt(0.5)/Co(0.5)]₈? The value of zero R_H does not fall on the H_z - R_H loop.

Reply: In Supplementary Figure 12, the Pt(3.5)/[Pt(0.5)/Co(0.5)]₈ sample has been set to demagnetization state before the SOT switching measurement. As a result of the

multidomain nature, the detected R_H is near zero.

c) Significant fields are measured by second harmonic measurements in [Pt(0.5)/Co(0.5)]₈. As probably switching would occur at some point (beyond 4×10^7 A/cm²) for this control sample, all I (mA) axes in the different no switching/switching loop figures could be converted in current density axes for improved clarity when comparing the different graphs. There is no reason to have either I or J instead of only J in the different graphs.

Reply: We agree with the reviewer that probably SOT switching could occur at some point beyond 4×10^7 A/cm² for the C2 control sample. Larger coercivity will generally need larger switching current density, which could also contribute to the absence of SOT switching in the C2 control sample. The result suggests that only interfacial asymmetry itself without the composition gradient can not lead to enough strong spin-orbit torque to switch the magnetization of a thick magnetic layer with a large coercivity within the maximum applied electrical current density of 4.5×10^7 A/cm².

In the revised manuscript, we have used current (I) rather than current density (J) for the current axes in different SOT switching loops. The reason is that current density is not suitable for the IrMn/Co/Ru/CoPt heterojunctions as different sublayers have different conductivity. In order to improve the clarity when comparing the different graphs, we have summarized both the total metal layer thickness and the critical switching current density at $H_x = +100$ Oe of all the studied samples in the revised supplementary table 1. In addition, both current and current density have been given in some related discussions.

2. Gradient scaling with thickness

1379-382 “Bulk spin-orbit torque in magnetic films [...] is independent of the magnetic layer thickness, and therefore the critical current density and the SOT effective magnetic field can be similar for different magnetic layer thickness”. While this can be true when the bulk SOT originates from structural asymmetry, I believe this statement to be misleading in the case of a gradient composition. The best achievable gradient reduces as $1/\text{thickness}$, and thus SOT effective magnetic field is limited by an amount scaling as $1/\text{thickness}$ as well (see, e.g., Liu et al., Phys. Rev. B 101, 220402(R) (2020)). This sentence needs to be corrected to clearly state this limit. Especially because, when stating “By contrast, the spin-orbit torque effective magnetic field based on spin Hall effect and/or interfacial Rashba effect strongly depends on the magnetic layer thickness”, it actually also reduces exactly as $1/\text{thickness}$.

Reply: To avoid any misunderstanding, we have deleted this whole paragraph and added some related discussion in the revised manuscript. The paper [Phys. Rev. B 101, 220402(R) (2020)] mentioned by the reviewer has been added as reference 12 in the revised manuscript. The related discussion can be found in page 8 of the revised manuscript. It reads:

“When the bulk spin-orbit coupling keeps constant and plays dominant role, in principle there is no limit to the thickness of switchable magnetic films. However, as the Co composition δ in $\text{Co}_\delta\text{Pt}_{1-\delta}$ (δ is at%) alloy film linearly increases from δ_1 to δ_2 with the magnetic layer thickness t , the Co composition gradient varies with the thickness as $(\delta_2-\delta_1)/t$. Thus, due to small composition gradient in thick single magnetic layer, the SOT effective magnetic field may become too small to switch the magnetization. Further experiments indicate that a single CoPt composition gradient magnetic film thicker than 7 nm can not be switched by the SOT effect. In fact, a thicker CoPt magnetic film can be obtained by periodically depositing the CoPt composition gradient layer.”

Additional comment:

The domain-wall (DW) motion depends indeed on the chirality of the DWs, the electrical current, and the magnetic field as reminded by the authors. They deduced left-handed chiral Néel walls in the multilayer Pt(0.7)/Co(0.3)/Pt(0.5)/Co(0.5)/Pt(0.3)/Co(1). Reversing the gradient of composition, this should reverse DMI and chirality of the DWs (at the same time that it reverses polarization sign of the spin currents). Out of curiosity, did the authors observe any clue of reversed DW chirality/DMI field sign in samples B1-B3 with opposite composition gradient? This would nicely correlate with the reversal of the SOT, on the phenomenological level.

Reply: It is an excellent suggestion. It is true that reversing the gradient of composition should reverse DMI and chirality of the DWs. Measurements shown in Reply-Figure 2 indicate that the DWs velocity of B2 sample with negative Co composition gradient shows opposite dependency on the external magnetic field compared to A0 sample with positive Co composition gradient. The velocity of the down-to-up DW in B2 sample ($V_{\text{DN-UP}} = 0$) becomes zero at $H_x = +70$ Oe. This means that the DMI effective magnetic field is -70 Oe for the down-to-up DW in B2 sample. So the down-to-up DW in B2 sample is the right-handed chiral Néel DW ($\downarrow\leftarrow\uparrow$). But the DMI effective magnetic field is +155 Oe for the down-to-up DW in A0 sample, i.e. the left-handed chiral Néel DW ($\downarrow\rightarrow\uparrow$). These results confirm the reversal of DMI field and DW chirality when reversing the composition gradient of the CoPt alloy films. The related discussion can be found in page 7 and page 12 of the revised manuscript. It reads:

“More interestingly, not only the polarity of magnetization switching, but also the DMI effective magnetic field, and the chirality of the DWs reverse when the composition gradient becomes opposite (it will be discussed below), highlighting the critical role played by the composition gradient in the magnetization switching.”

“By contrast, we found that CoPt films with negative Co composition gradient has the right-handed chiral Néel DW and the DMI effective magnetic field in the right-handed down-to-up DW is -70 Oe, as shown in Supplementary Fig. 10e.”

Reply-Figure 2. Domain wall velocity versus in-plane magnetic field H_x for B2 sample [Pt0.5/Co0.5/Pt0.6/Co0.4/Pt0.7/Co0.3]. Red and blue symbols represent up-to-down (UP-DN) and down-to-up (DN-UP) DWs, respectively. Square and circular symbols correspond to positive and negative currents, respectively.

Reviewer #2 (Remarks to the Author):

Review of manuscript resubmitted to nature communications

Review of corrections after previous response

This is a review of the modified manuscript, previously reviewed for a different Nature journal.

Since previous reviews are on-record, I refrain from the standard full review of the paper's pros and cons.

In a nutshell, this paper is in the field of SOT switching of PMA thin films.

The 2 main claims are that 1) A magnetic "composition gradient" layer has broken inversion symmetry and thus enables SOT to arise directly in a PMA layer. 2) Demonstration of field free switching by coupling the PMA layer to a stable Co layer with in-plane magnetization.

Overall, the authors have done a good job incorporating previous comments from the reviewers, which has improved their paper.

To re-state my previous assessment. The ideas presented by the authors, of the SOT switching in a magnetic gradient layer are worthy of publication in Nature communications. The results support the claims in a satisfactory manner. This area of research is of interest to the scientific community and is also of relevance for applications. So, while the figure of merit is not (much) better than previously published results, bringing these concepts to the community is important, and maybe future work will also increase the FOM.

The second concept of field free switching is not related to the first. It is a proof-of-concept that it is possible to replace the external field with an antiferromagnetically coupled in-plane magnetic layer. It is demonstrated on the gradient layer, and the data and analysis are convincing.

Thus I support publication of this paper, after addressing a few final comments.

Reply: We really appreciate the reviewer for the positive recommendations.

First, the readability of the article has reduced. A good scientific editing is in place before publication.

Reply: The English has been further polished to improve the readability of the revised manuscript.

An issue that I would like to have stated for the record – the EDS data of the TEM is always noisy, so there is always some spread of the results. I don't think that the gradient is a continuous gradient like the authors claim, but it is more of set-gradient that after EDS spread appears as a gradient. This does not change the results, but I would predict that a true gradient that is prepared by continuous deposition could have a different outcome.

Reply: We agree with the reviewer about the EDS data and composition gradient. We have accepted the useful suggestion of reviewer and added it in pages 5&6 of the revised manuscript. It reads:

“However, due to the spread of EDS data, the variation of the composition in our designed samples may be not as continuous as the EDS data, and a more continuous composition gradient can be prepared by continuous deposition.”.

Regarding the field free switching. This is an interface effect. So its usefulness will be reduced when a thicker FM layer is used, and thus it may be limited to thin layers. The authors should estimate what is the maximum thickness of the gradient layer (from their anisotropy energy analysis of the layer) that field free will work. Will it work for the 9nm film?

Reply: We agree with the reviewer that the field free switching will be difficult to achieve when a thicker FM layer is used. Considering the fact that the remarkable SOT switching in our CoPt films can be observed at the external magnetic field of 100 Oe, the minimum interlayer exchange coupling field should be bigger than 100 Oe to achieve the field free switching of the maximum magnetic layer thickness. In a simple estimation, we obtain the interlayer exchange coupling field $H_{IEC} = J/M_S t_{FM}$, where J is the interlayer exchange coupling constant, M_S is the saturation magnetization and t_{FM} is the thickness of FM layer. According to $H_{IEC} = 550$ Oe in the studied IrMn/Co/Ru/CoPt with the CoPt layer thickness $t_{FM} = 3.3$ nm and $M_S \approx 250$ emu/cm³, we get $J = 0.0454$ erg/cm². Assuming that the interlayer exchange coupling constant $J = 0.0454$ erg/cm² keeps unchangeable, and a thick magnetic layer has saturation magnetization $M_S = 500$ emu/cm³ and $H_{IEC} = 100$ Oe, we can get the maximum magnetic layer thickness $t_{FM} = 9$ nm. In fact, if we choose a proper Ru layer thickness, we can obtain much big interlayer exchange coupling constant to realize the field free switching of a relative thick magnetic layer. For most applications in spintronics devices the magnetic layer thickness is usually less than 10 nm. In this sense, field free switching realized by using interlayer exchange coupling is promising.

The corresponding revision can be found in the last paragraph of page 18 in the

revised manuscript. It reads:

“First, field-free bulk SOT switching of CoPt composition gradient film is demonstrated for the first time in IrMn/Co/Ru/CoPt heterojunctions, relying on synthetic antiferromagnetic interlayer coupling through a Ru spacer to an auxiliary Co layer exchange-biased by IrMn. Considering the fact that a remarkable SOT switching in our CoPt films can be observed at the external magnetic field of 100 Oe, the minimum interlayer exchange coupling field should be bigger than 100 Oe to achieve the field-free switching of the maximum magnetic layer thickness. In fact, if we choose a proper Ru layer thickness, we can obtain much big interlayer exchange coupling constant to realize the field-free switching of a relative thick magnetic layer.”

The results for the 9nm film are a bit strange, looking at the table in the supplementary. E.g., why is the theta so much larger for A4? Is it just because the thicker layer causes more of the current to run in the FM and not the Ru buffer layer? If so, then why is the critical switching current density not smaller? This actually is more reasonable, as the coercivity fields were larger, and the critical current was expected to also be larger in this case. Please comment.

Reply: In Supplementary Table 1, we can see that the inset of 0.3 nm MgO in the 9.9 nm CoPt film (A4 sample in Supplementary Table 1) simultaneously enhance the coercivity and spin Hall angle as compared with the 3.3 nm CoPt film (A0 sample in Supplementary Table 1), and as a result they have similar critical switching current density.

First, we agree with the reviewer that the coercivity of A4 sample is larger and the critical current is expected to also be larger. In addition to enhancing the perpendicular magnetic anisotropy, here MgO can be regarded as “pinning defect” in magnetic film to enhance the coercivity. Second, the existence of MgO in the A4 sample could introduce additional interface Rashba effect. Since this interface Rashba effect induced symmetry-broken is also along the film growth direction, it would enhance the composition gradient induced bulk symmetry-broken. As a result, A4 sample shows a larger effective spin Hall angle. Finally, the critical switching current density is determined by the competition between at least two factors: the increase of coercivity leads to the increase of critical switching current density; and the enhanced SOT switching leads to the decrease of critical switching current density. We have added some comments in page 20 of the revised manuscript. It reads:

“Now let us discuss the relation between the composition gradient and the bulk SOT effect. The composition gradient itself can be greatly modulated by varying the layer thickness and layer number of Co and Pt layers, and correspondingly the SOT effective magnetic field can be effectively tuned as summarized in Supplementary Table 1. Qualitatively speaking, the spin Hall angle, the DMI effective magnetic field, the

chirality of the DWs, the damping-like effective magnetic field, and the polarity of magnetization switching all reverses when the composition gradient becomes opposite. But it is hard to quantitatively compare these physical parameters between samples with different composition gradient. For example, the inset of 0.3 nm MgO in the 9.9 nm CoPt film (A4 sample in Supplementary Table 1) simultaneously enhance the coercivity and spin Hall angle as compared with the 3.3 nm CoPt film (A0 sample in Supplementary Table 1), and as a result they have similar critical switching current density.”

Reviewer #3 (Remarks to the Author):

With the three rounds of the review process in Nature Electronics, the authors have made substantial revisions to this manuscript by conducting new experiments, improving the physics model, and implementing all the referees' comments. In addition, my comments in the last round of the review process have been carefully addressed and implemented in the manuscript accordingly. Although a quantitative model of bulk SOT remains to be established, its qualitative version interpreting the relation of bulk-SOT and composition gradient has been provided in accord with systematic experiments. I agree with the other referees that at this stage, nothing would affect the novelty of this work or prevent its publication.

As a fundamental study, this work will stimulate many further research proposals on this new bulk-SOT discovered in a magnetic system with a composition gradient and will help the researchers design more efficient magnetic storage. In my opinion, the current manuscript is already in a good shape in terms of convincing conclusions and clear statements of novelty and potential applications. Hence, I recommend this paper to be accepted in Nature Communications without major changes.

Reply: We are grateful to the reviewer for supporting the publication of this work in Nature Communications.

Reviewers' Comments:

Reviewer #1:

Remarks to the Author:

The authors have addressed satisfactorily the last comments made by the referees and the manuscript now appears ready for publication.

Reviewer #2:

Remarks to the Author:

I recommend publication of this article in Nature communications, and do not have any new scientific criticism or concerns to the authors. The conclusions are well supported by the experiments.

Some of the results, which the referees discussed with the authors, do not have a full quantitative theory. This is completely acceptable, as the novel claims are supported by the results. These details will be addressed by the authors and other groups who decide to continue this line of research after reading this paper.

I commend the authors on their efforts of improving the manuscript (considerably) by taking under consideration all our comments at each and every review cycle; performing additional experiments and calculations and introducing the corrections to the manuscript version.

REVIEWERS' COMMENTS

Reviewer #1 (Remarks to the Author):

The authors have addressed satisfactorily the last comments made by the referees and the manuscript now appears ready for publication.

Reply: We are grateful to the reviewer for supporting the publication of this work in Nature Communications.

Reviewer #2 (Remarks to the Author):

I recommend publication of this article in Nature communications, and do not have any new scientific criticism or concerns to the authors. The conclusions are well supported by the experiments.

Some of the results, which the referees discussed with the authors, do not have a full quantitative theory. This is completely acceptable, as the novel claims are supported by the results. These details will be addressed by the authors and other groups who decide to continue this line of research after reading this paper.

I commend the authors on their efforts of improving the manuscript (considerably) by taking under consideration all our comments at each and every review cycle; performing additional experiments and calculations and introducing the corrections to the manuscript version.

Reply: We would like to thank the reviewer for supporting the publication of this work in Nature Communications.

Again, we sincerely appreciate all the referees for their valuable comments and suggestions that greatly helped us to improve the quality of this article.